# White Adipose Tissue as a Site for Islet Transplantation

**Naoaki Sakata [1,2,*]**, **Gumpei Yoshimatsu [1,2]** and **Shohta Kodama [1,2]**

[1]    Department of Regenerative Medicine and Transplantation, Faculty of Medicine, Fukuoka University,
7-45-1 Nanakuma, Jonan, Fukuoka 814-0180, Japan; gyoshimatsu@fukuoka-u.ac.jp (G.Y.);
skodama@fukuoka-u.ac.jp (S.K.)

[2]    Center for Regenerative Medicine, Fukuoka University Hospital, 7-45-1 Nanakuma, Jonan,
Fukuoka 814-0180, Japan

*    Correspondence: naoakisakata@fukuoka-u.ac.jp; Tel.: +81-92-801-1011 (ext. 3631)

**Abstract:** Although islet transplantation is recognized as a useful cellular replacement therapy for severe diabetes, surgeons face difficulties in islet engraftment. The transplant site is a pivotal factor that influences the engraftment. Although the liver is the current representative site for clinical islet transplantation, it is not the best site because of limitations in immunity, inflammation, and hypoxia. White adipose tissue, including omentum, is recognized as a useful candidate site for islet transplantation. Its effectiveness has been evaluated in not only various basic and translational studies using small and large animals but also in some recent clinical trials. In this review, we attempt to shed light on the characteristics and usefulness of white adipose tissue as a transplant site for islets.

**Keywords:** white adipose tissue; islet transplantation; omentum; epididymal fat pad; adipose tissue-derived stem cell; adiponectin

## 1. Introduction

Islet transplantation (ITx) is considered as a promising and reliable cellular replacement therapy for severe diabetes mellitus (DM) patients with unstable condition of blood glucose (BG) level despite intensive insulin therapy, especially for insulin-dependent type 1 DM patients. The therapeutic outcomes of ITx have gradually, but dramatically, improved through innovations in technology regarding islet isolation, transplantation procedures, and immunosuppressants [1]. The most recent report from the Collaborative Islet Transplant Registry indicated that the insulin-independence (no necessity to use daily insulin injections) rate at 3 years after ITx was 44% [2], and that approximately 80% of the recipients who received 600,000 or more total islet equivalents (IEQs) achieved insulin independence [3]. Furthermore, a phase III study for elucidating the therapeutic effects of clinical ITx in type 1 DM (CIT-07) performed at eight centers in the United States until 2017 revealed that 87.5% and 71% of the diabetic participants achieved an HbA1c level of less than 7.0% and prevention of severe hypoglycemic events at 1 and 2 years after the first ITx, respectively [4]. Although ITx is recognized as a useful therapy that enables an appropriate physiological supply of insulin responding to the changes of blood glucose levels and prevents severe hypoglycemia and life-threatening complications related to micro- and macroangiopathy, including cardiomyopathy, nephropathy, retinopathy, and neuropathy, it still involves some problems that compromise the therapeutic effects.

One of the problems surgeons face is the unsatisfactory transplant efficacy, which depends on the difficulty in engraftment of transplanted islets. Many transplanted islets fail to engraft in a couple days after transplantation [5] because they suffer from harsh environmental factors of immunity [6], inflammation triggered by the innate immune system [7], and ischemia [5], which are affected by the transplant site. For instance, the liver is a major clinical transplant site for islets. However, the liver is not the best site in terms of immunity (owing to liver-resident macrophage (Kupffer cells) and natural

killer cells [8–10]), inflammation (instant blood-mediated inflammatory reaction [IBMIR], an acute thrombotic and inflammatory reaction that causes damage to transplanted islets [11]), and hypoxia (owing to embolization of the peripheral portal vein by the transplanted islets themselves [5,12]). To date, various organs, including the renal subcapsular space [13], gastrointestinal tract [14], bone marrow [15], spleen [6,16], and muscle and subcutaneous tissue [17], have been examined to assess their characteristics as alternative sites for islets in an effort to establish the ideal transplant site (Table 1). Though they offer various attractive advantages, all of these sites have also limitations, which become obstacles for use in the clinical setting [6,17,18].

**Table 1.** Candidates for transplant site for islets and their characteristics.

| | Advantages | Disadvantages |
|---|---|---|
| Liver | ✓ Representative transplant site for clinical islet transplantation<br>✓ Largest abdominal organ, which enables the storage of a high volume of islets<br>✓ Physiological insulin delivery<br>✓ Comparatively little invasion in the transplant procedure | ✓ Immunity<br>✓ IBMIR<br>✓ Difficulty in monitoring<br>✓ Risk of portal thrombosis and hypertension |
| Kidney | ✓ Preventing direct contact of blood flow (diminishing the risk of IBMIR)<br>✓ Best transplant efficacy in small animal studies | ✓ Difficulty of transplant procedure in clinical<br>✓ Systemic insulin delivery |
| Spleen | ✓ Rich vascularity<br>✓ Physiological insulin delivery<br>✓ Regulation of immunity<br>✓ Islet regeneration | ✓ Risk of IBMIR<br>✓ Risk of bleeding following transplant procedure |
| Muscle/ subcutaneous tissue | ✓ Minimized invasion<br>✓ Safety<br>✓ Preventing risk of IBMIR | ✓ Hypoxia<br>✓ Hypovascularity<br>✓ Immunity |
| Omentum (white adipose tissue) | ✓ Physiological insulin delivery<br>✓ Capacity to involve a large number of islets (omental pouch) | ✓ Possibility of surgical complications, including adhesion ileus |
| Mesentery (white adipose tissue) | ✓ Physiological insulin delivery | ✓ Necessity for bowel resection in the case of graft removal |

IBMIR, instant blood-mediated inflammatory reaction.

In this review, we attempted to shed light on the characteristics and usefulness of white adipose tissue as a transplant site for islets. White adipose tissue, such as omentum (i.e., omental pouch) and mesentery, has the advantages of a high capacity for accommodating a high volume of islets and of belonging to the portal venous system, which provides physiological insulin delivery (Table 1), and is considered a useful candidate as a transplant site of islets. It is believed that these characteristics might be suitable for the clinical setting, especially when the liver cannot be used owing to portal thrombosis and hypertension.

## 2. Previous Trials Using White Adipose Tissue as a Transplant Site for Islet Transplantation

White adipose tissue is a connective tissue that contains mainly lipids and many small vessels. It is found in subcutaneous tissue, abdominal cavity (e.g., omentum and mesentery), bone marrow, muscle, and the breast. Of note, it is believed that white adipose tissue has mainly two advantages as a transplant site for ITx as follows: its spatial capacity and prevention of direct contact between islets and blood. The former is a preferable characteristic for ITx, which enables the accommodation of high tissue volume. The latter contributes to the prevention of IBMIR, which impairs the engraftment of transplanted islets and leads to complications, including bleeding and thrombosis. Because of these characteristics, white adipose tissue has been used as the site for implantation of bioscaffolds, which support the engraftment of islets or encapsulated islets that cannot be infused via vessels. The encapsulation technique is used to protect transplanted islets from the recipient's immune system (Figure 1).

The first ITx study using white adipose tissue was published in the early 1980s. Outzen and Leiter performed syngeneic and allogeneic rodent ITx into the mammary fat pad and assessed its therapeutic effect and engraftment [19]. To date, various studies have been performed using mainly the three following kinds of white adipose tissues: omentum, mesentery, and epididymal fat pad.

**Figure 1.** Characteristics of white adipose tissue as the transplant site for islets.

Omentum has been well examined as a candidate islet transplant site. The greater omentum is a large visceral peritoneum that extends from the greater curvature of the stomach, passing in front of the small intestines, and doubling back to ascend to the transverse colon before reaching the posterior abdominal wall. The greater omentum plays a key role in not only peritoneal immunity but also storage of adipose tissue. The omental pouch is a major target of islet transplant site in the omentum. The omental pouch is defined as a space consisting of the greater omentum, which enables the transplantation of a high volume of cells or tissues. Another merit of the omentum is in the physiological drainage route of insulin, because the vascular system of the greater omentum belongs to the portal system, as mentioned in the Introduction (Table 1, Figure 1) [20]. The usefulness of the omentum as an islet transplant site was first revealed in 1983 [21]. Since then, various groups have examined its effectiveness in clinical studies (Table 2) [22–39]. For example, Tuch et al. transplanted allogeneic fetal pancreatic tissues into the omental pouch and muscle of a 29-year-old insulin-dependent diabetic woman in the mid-1980s. Although this was the first clinical case of allogeneic islet transplantation into the omental pouch, there was no engraftment of β cells and no detection of plasma C-peptide at 13 months after transplantation, despite the use of immunosuppressants [40]. In early studies of the

omentum, its usefulness was assessed using large animals such as dogs and monkeys [22,34,35,38]. Regarding rodents, rats are mainly used for basic studies. Although the mouse is a major animal model in general, the greater omentum of mice is too small to use for ITx studies [41]. Instead, the epididymal fat pad is popular for ITx studies using mouse models. The focus of recent studies using omentum is to assess the usefulness of bioscaffold, which supports islet engraftment and encapsulated islets that cannot be infused intravenously. In particular, bioscaffolds are used to induce prevascularization in the omentum, which protects transplanted islets from hypoxia resulting from ischemia. Kriz et al. developed a vascularized pocket using the omental pouch by preimplantation of a polymer spacer. They succeeded in achieving normoglycemia of diabetic rats by transplantation of syngeneic islets into the prevascularized pocket [30,39]. Pedraza et al. evaluated the characteristics of a microporous bioscaffold by implantation with islets into the omental pouch of diabetic rats [29]. As in other trials, growth factors such as vascular endothelial growth factor [26,27,39] and angiogenesis-promoting cells such as mesenchymal stromal cells (MSCs) [27] and endothelial cells [31] were used for vascularization of transplanted islets into the omentum. Regarding encapsulated islets, agarose gel (Kobayashi et al.), alginate gel (Pareta et al.), and alginate gel (Ibarra et al.) were assessed to determine the therapeutic effect of implantation into the omental pouch [25,28,36]. The encapsulation technique is aimed at rendering protection, including islets in biomaterials, from the immune response of the recipient.

**Table 2.** Representative studies regarding the omental pouch.

| Author (Year) | Transplant Model (Animal) | Number of Transplanted Islets | Additional Treatment | Outcome | Reference |
|---|---|---|---|---|---|
| Kasoju (2020) | Syngeneic (rat) | Not described | Using biomaterial spacer and growth factor | Islet engraftment | [39] |
| Lu (2019) | Not described | 450–500 islets | Using hydrogel | Normoglycemia achieved immediately | [24] |
| Ibarra (2016) | Allogeneic (rat) | 800–1000 beads (accurate number not described) | Encapsulation | Transplant efficacy was unclear | [25] |
| Montazeri (2016) | Xenogeneic (rat to nude mouse) | 250 islets (IEQs) | Using oxygenation technique with growth factor | Improved blood glucose level | [26] |
| Hajizadeh-Saffar (2015) | Allogeneic (mouse) | 200 islets (IEQs) 400 islets (IEQs) | Co-transplantation with growth factor-releasing cells (derived from mesenchymal stromal cells) | Normoglycemia rates were 80% in 200 IEQs with MSCs and 40% in 400 IEQs | [27] |
| Pareta (2014) | Syngeneic Allogeneic (rat) | 800 islets | Encapsulation | Failed to achieve normoglycemia | [28] |
| Pedraza (2013) | Syngeneic (rat) | 1800 islets | With or without bioscaffold | Achieved normoglycemia over 110 days | [29] |
| Kriz (2012) | Syngeneic (rat) | 10,000 islets (IEQs)/kg (2000–3000 IEQs) | Using biomaterial spacer | Normoglycemia rate was 70% at 100 days after transplantation | [30] |
| Gupta (2011) | Syngeneic Allogeneic (rat) | 2000 islets | Using endothelialized modules | Normoglycemia rate was 40% (syngeneic) | [31] |
| Berman (2009) | Autologous Allogeneic (monkey) | 5093 IEQ/kg (autologous) 4200–14,544 IEQ/kg (allogeneic) | Bioscaffold Immunosuppressants (allogeneic) | Achieved normoglycemia in autologous islet transplantation. Therapeutic effect was similar to that of intrahepatic islet transplantation | [34] |
| Kobayashi (2006) | Syngeneic (mouse) | 1500 islets | Encapsulation | The normoglycemia rate was 90% over 100 days after transplantation | [36] |
| Kin (2003) | Syngeneic (rat) | 2000 islets | No additional treatment | Achieving normoglycemia at 56 days after transplantation | [23] |
| Guan (1998) | Syngeneic (rat) | Approximately 3000 islets | No additional treatment | Achieving normoglycemia at 2 months after transplantation | [37] |

IEQ, islet equivalent.

The advantage of the mesentery as a transplant site for islets is its large space and physiological insulin delivery, much like the omentum. The mesentery belongs to the portal venous system, which contributes to achieving physiological insulin delivery, and it is able to accommodate high volumes of islets (Figure 1). On the other hand, it presents challenges in terms of graft removal without sacrificing the intestinal tract [42]. To the best of our knowledge, there have been few studies that have assessed the characteristics and usefulness of the mesentery as an islet transplant site. The groups led by

Rajotte and Michalska demonstrated the usefulness of the mesentery in the 1990s and early 2000s [43,44]. Later, Weaver et al. evaluated the islet transplant efficacy of various white adipose tissues, including the subcutaneous tissue, mesentery, and epididymal fat pad. They showed that the normoglycemia rate of islet-transplanted mice (600 IEQs) into the mesentery was superior to that into subcutaneous tissue but inferior to that into the epididymal fat pad. Additionally, some studies regarding implantation with bioscaffolds for immune isolation [45] and promotion of vascularization [46–48] have been reported.

The epididymal fat pad is also used as a transplant site ITx. It is defined as white adipose tissue in the perigonadal region, is considered the major visceral white adipose tissue in rodents [41], and is recognized as one of the largest visceral white adipose tissues, whereas the greater omentum of rodents, especially that of mice, is too small, as previously described. Although the epididymal fat pad has the disadvantage of systemic insulin release, it has been used for the experimental ITx studies instead of the omentum because of its ease in operability (Figure 1). Many studies have shown its usefulness in assessing various types of bioscaffold, including elastin [49], fibrin [50], poly(lactide-co-glycolide) [51–54], polyethylene glycol [13,42,55–57], alginate (ALG) [58,59], and poly-l-lysine [60], much like the omentum (Table 3).

**Table 3.** Representative studies regarding the epididymal fat pad.

| Author (Year) | Transplant Model | Number of Transplanted Islets | Additional Treatment | Outcome | Reference |
|---|---|---|---|---|---|
| Nguyen (2020) | Xenogeneic (rat to nude mouse) | 500 islets (IEQs) | With or without bioscaffold | Achieved normoglycemia (88% with bioscaffold, 44% without bioscaffold) | [61] |
| Minardi (2019) | Syngeneic (mouse) | 70 islets | With bioscaffold | Achieved normoglycemia at 1 month after transplantation | [49] |
| Liu (2018) | Syngeneic Allogeneic (mouse) | 250 islets | Using immunomodulation technique and bioscaffolds | Achieved normoglycemia in all mice | [51] |
| Manzoli (2018) | Allogeneic (mouse) | 750–1000 islets (IEQs) | Encapsulation | Achieved normoglycemia at 10 days after transplantation | [58] |
| Weaver (2017) | Syngeneic (mice) | 600 islets (IEQs) | Using bioscaffold with growth factor | The normoglycemia rates were 75% and 60% using bioscaffold with and without growth factor, respectively | [42] |
| Wang (2017) | Syngeneic (mouse) | 150–500 islets | Using bioscaffold | Mice achieved normoglycemia: 10/12 (500 islets + scaffold), 10/15 (250 islets + scaffold), 9/19 (150 islets + scaffold), 3/10 (250 islets) | [62,63] |
| Mao (2017) | Syngeneic (mouse) | 300 islets | Using bioscaffold | No mice achieved normoglycemia in ITx only. All mice achieved normoglycemia with bioscaffold | [64] |
| Buitinga (2017) | Syngeneic (mouse) | 300 islets | Using bioscaffold | Achieved normoglycemia (75% with bioscaffold, 29% without bioscaffold) | [65] |
| Villa (2017) | Allogeneic (mouse) Xeneogeneic (baboon to NOD/scid mouse) | 750 islets (IEQs) | Encapsulation | Allogeneic: Normoglycemia achieved and maintained in all the mice (7/7) for 100 days after transplantation Xenogeneic: Normoglycemia achieved and maintained in all the mice (4/4) for 30 days after transplantation | [59] |
| Rios (2016) | Syngeneic (mouse) | 300 and 500 islets | Using bioscaffold | Achieved normoglycemia (100% in 500 islets, 25% in 300 islets) | [55] |
| Liu (2016) | Syngeneic (mouse) | 250 islets | Using bioscaffold | Normoglycemia achieved and maintained for 80 days after transplantation | [52] |
| Najjar (2015) | Syngeneic (mouse) | 250 islets (IEQs) | Using bioscaffold | Achieved normoglycemia (60% with bioscaffold, 10% without bioscaffold) | [50] |
| Gibly (2013) | Xenogeneic (Human to NOD/scid mice) | 2000 islets (IEQs) | Using bioscaffold | Normoglycemia achieved and maintained over 140 days after transplantation | [66] |
| Brady (2013) | Syngeneic (mouse) | 250 islets (IEQs) | Using bioscaffold | Achieved normoglycemia (100% with bioscaffold, 87.5% without bioscaffold) | [67] |
| Gibly (2011) | Syngeneic (mouse) | 75 islets | Using bioscaffold | Normoglycemia achieved and maintained for 42 days after transplantation | [53] |
| Kheradmand (2011) | Allogeneic (mouse) | 500 islets | Using bioscaffold with splenocytes | Achieved normoglycemia for 150 days (80% with splenocytes) | [54] |
| Brubaker (2010) | Syngeneic (mouse) | 150 islets | Using bioscaffold | Normoglycemia achieved and maintained for 110 days after transplantation (both with and without bioscaffold) | [56] |
| Salvay (2008) | Syngeneic (mouse) | 125 islets | Using bioscaffold | Normoglycemia achieved and maintained for 300 days after transplantation | [57] |

Unlike in rodents, dogs, and monkeys, subcutaneous tissue in humans is composed of connective tissue and white adipose tissue. In other words, subcutaneous tissue can be defined as one of the white adipose tissues in humans. Although subcutaneous tissue has the merit of easiness and safety with minimum invasion of the transplant procedure and monitoring the condition of islet graft, it harbors the poorest transplant efficacy owing to hypoxia, hypovascularity, and immunity (Figure 1) [17]. Conversely, if hypoxic and hypovascular condition can be removed (e.g., prevascularization and hyperbaric oxygenation [68]), subcutaneous tissue can be a promising site for islets. We previously reviewed subcutaneous islet transplantation [17] and thus, show the recent topics about this area in this review. Recently, subcutaneous white adipose tissue in the inguinal area was assessed in a new trial of ITx using white adipose tissue. In general, subcutaneous tissue requires prevascularization for success in ITx because of its hypovascular environment. For example, Forster et al. performed prevascularization of subcutaneous white adipose tissue in the inguinal area using a silicone spacer with Matrigel® and a growth factor and then evaluated the effect of ITx into the spacer [69]. Yasunami et al. performed ITx into the subcutaneous white adipose tissue in the inguinal area without prevascularization and revealed that the transplant efficacy was superior to that of intrahepatic ITx. Furthermore, they also successfully performed allogeneic ITx with 4 months of normoglycemia under immunosuppressants using this site [70]. They showed that it is possible to use subcutaneous tissue as a transplant site for islets with promising therapeutic effects by selecting a preferable site.

The transplant efficacy of ITx using white adipose tissue is unclear. However, some groups have successfully achieved normoglycemia using the epididymal fat pad by one donor [49,53,56] and others by more than two donors [42,50,52,55,62–65,67]. In our experience, 200 islets can be obtained from a single mouse. It is estimated that the transplant efficacy of the epididymal fat pad might be inferior to that of the kidney, but superior to that of the liver. Furthermore, the transplant efficacy might depend on the white adipose tissue. Weaver et al. compared the transplant efficacy of the epididymal fat pad, mesentery, and subcutaneous white adipose tissue and revealed that the epididymal fat pad was the best transplant site among the white adipose tissues [42]. On the other hand, other groups have shown that the transplant efficacy using the omentum was similar to that of the epididymal fat pad [32].

## 3. Characteristics of White Adipose Tissue as a Site for Islet Transplantation

Although previous studies have revealed the usefulness of white adipose tissue as a transplant site for ITx, the mechanism that enables the engraftment of the transplanted islets in the white adipose tissue has not been fully discussed. We discuss the mechanism from the views of extracellular matrix (ECM) and cellular components of white adipose tissue.

### 3.1. ECM of White Adipose Tissue

White adipose tissue is a connective tissue composed of various ECMs, including collagen types I, IV, V, and VI, laminin, and fibronectin [71–73]. It is believed that these ECMs support the adhesion of transplanted islets into the white adipose tissue via integrin.

Integrin is one of the adhesion factors expressed on the surface of cellular membranes [74]. It is a heterodimer composed of a combination of 18 types of α subunits (α1–11, E, V, L, M, X, D, and IIb) and 8 types of β subunits (β1–8) that works as a receptor for ECMs. For instance, integrins α1β1, α2β1, α10β1, and α11β1 work as primary receptors for collagens [75]. Integrin α3β1 also mediates attachment to collagen [76]. Regarding other ECMs, integrins α1β1, α2β1, α3β1, α6β1, α7β1, and α6β4 are the receptors for laminin [77], and integrins α3β1, α4β1, α5β1, α8β1, α9β1, αvβ1, αIIbβ3, αvβ3, αvβ6, and α4β7 are receptors for fibronectin [78,79]. On the other hand, it has been shown that islets harbor at least five integrin heterodimers, including α1β1, α3β1, α5β1, αvβ1, and α6β1 [80]. In other words, islets can attach to white adipose tissue via a combination of collagen, laminin, fibronectin, and integrin, which may contribute to the engraftment of islets to white adipose tissue. Furthermore, these correlations between ECMs and adhesion molecules may also support the improvement of the function of engrafted islets. Kaido et al. revealed that the insulin-releasing

function of cultured human islet β cells was improved by attachment to collagen IV via integrin α1β1 expressed on the β cells [81]. Bosco et al. also showed that a combination of laminin and integrin α6β1 enhanced the insulin-releasing function of rat islets in glucose-stimulated insulin secretion [82]. Furthermore, Wang's group clarified the importance of integrin β1 in islets in the development of fetal pancreas and prevention of islet-cell apoptosis via activation of the FAK, MAPK, and ERK signaling pathways [83–85]. They later showed integrin α3-regulated islet-cell survival and function via the PI3K and Akt signaling pathway [86].

Recently, we showed that the transplant efficacy of "fat-covered" ITx using epididymal white adipose tissue was superior to that of intraperitoneal ITx and nearly equal to that of renal subcapsular ITx in a rodent model. This "fat-covered" transplantation method is very simple. The epididymal white adipose tissue was mobilized outside of the peritoneal cavity and distended under general anesthesia. Islets were dropped onto the white adipose tissue using a micropipette and covered. Suturing of the adipose tissue was not performed and bioscaffolds and biobinding agents (i.e., ECM) for binding islets to the white adipose tissue were not used. In this study, we showed that fibronectin was expressed in the epididymal adipose tissue and that islets expressed integrin β1, a receptor of fibronectin. It was estimated that transplanted islets were attached to epididymal white adipose tissue via a combination of fibronectin and integrin β1 and, therefore, engraftment of the islets was promoted (Sakata et al., Transplantation 2020, doi:10.1097/TP.0000000000003400; in press). However, the role of adhesion factors in the engraftment of islets into white adipose tissue has not been fully discussed. However, further studies are necessary.

### 3.2. Cellular Components of White Adipose Tissue

White adipose tissue consists of various cellular components, including adipocytes, adipose tissue-derived mesenchymal stromal (or stem) cells (ADMSCs), vascular endothelial cells, pericytes, macrophages, and different types of white blood cells. Among them, ADMSCs and adipocytes are the main cellular components of white adipose tissue that might contribute to the engraftment of islets.

### 3.2.1. ADMSCs

Although ADMSCs represent only 5% of the cellular components of white adipose tissue, they play important roles in tissue repair and immune modulation, as do other MSCs in other organs [87,88]. ADMSCs have received attention as cellular candidates for co-transplantation with islets, and many groups have examined their therapeutic effects. The functions of cotransplanted ADMSCs, which might promote the engraftment of islets, are classified into the following three types: neovascularization, reduction of inflammation, and regulation of immunity. At first, ADMSCs promote the formation of a neovascular network between the transplanted islets and recipient by secretion of various growth factors, including vascular endothelial growth factor [89,90], hepatocyte growth factor [89], and transforming growth factor-β [91]. For instance, Ren et al. showed that co-transplantation of ADMSCs increased islet revascularization by induction of hepatocyte growth factor and angiopoietin-1 expressions [92]. Furthermore, ADMSCs reduce proinflammatory cytokines such as tumor necrosis factor (TNF)-α [93,94], interferon-γ [91], interleukin (IL)-6β [93], and IL-17 [91]. Regarding immunomodulation, ADMSCs support immunotolerance in the transplant site by hindering the infiltration of CD4+ and CD8+ T cells [95] and macrophages [94], as well as by promoting the production and infiltration of Tregs [96]. A recent study revealed that xenogeneic co-transplantation of human ADMSCs with neonatal porcine islets led to earlier achievement normoglycemia in diabetic mice compared with transplantation of porcine islets only [97]. These functions of ADMSCs are characterized as supportive effects for promoting the engraftment of transplanted islets. On the other hand, transplanted ADMSCs can ameliorate the diabetic condition by enhancing the proliferation of transplanted islet cells [98], by differentiation into insulin-producing cells themselves [99,100], and by promoting insulin-releasing function of cotransplanted islets [91,93].

Although many studies on the therapeutic effect of ADMSC transplantation have been published, it is still unclear whether resident ADMSCs support the engraftment of transplanted islets into white adipose tissue because of the difficulty of proving their therapeutic effect with the current experimental models. Furthermore, the population of ADMSCs in white adipose tissue is too small, whereas adipocytes represent approximately 90% of white adipose tissue [101]. That resident ADMSCs contributing to the engraftment of transplanted islets offer no doubt. However, their therapeutic effect might be limited because of the smaller population of ADMSCs in white adipose tissue.

### 3.2.2. Adipocytes

As described previously, adipocytes are the major components of white adipose tissue. They are histologically seen as the cells that contain a large lipid droplet in their cytoplasm. Adipocytes play an important role in lipid regulation and cytokine ("adipocytokines") release [102]. Regarding lipid regulation, adipocytes store free fatty acids (FFAs) and esterify into triglycerides in energy excess and turn triglycerides into FFAs by using contained enzymes, including adipose triglyceride lipase in a decrease of energy. FFAs work for insulin resistance. Recently, adipocytes became known as the cells producing various adipocytokines. For example, the expression of TNF-$\alpha$, which induces inflammation and apoptosis, was seen in the white adipose tissue of obese mice [103]. Production of TNF-$\alpha$ in adipocytes is increased in type 2 diabetes [103,104]. Resistin and IL-6 are also produced in white adipose tissue and cause systemic inflammation and induce insulin resistance, similarly to TNF-$\alpha$ and FFA [105]. Monocyte chemoattractant protein-1 (MCP-1) is one of the adipocytokines that induces inflammatory reaction and insulin resistance via infiltration of macrophages in obesity [106–108]. It is not clear whether these adipocytokines promote islet engraftment. However, it is well known that TNF-$\alpha$ induced in the peritransplant period damages transplanted islets [7]. Furthermore, previous studies have shown that MCP-1 plays a pivotal role as an inflammatory cytokine in the early graft loss of islets [109,110]. On the other hand, it is difficult to evaluate the effect of IL-6 on transplanted islets. Min et al. revealed both beneficial and harmful effects of IL-6 blockade in ITx. Although blockade of IL-6 reduced innate inflammation after ITx, it inhibited revascularization of transplanted islets [111].

The usefulness for ITx has been shown in two adipocytokines as follows: leptin and adiponectin. Leptin is one of the representative adipocytokines that controls body weight by regulating food intake [112]. Regarding the therapeutic effect in ITx, Denroche et al. demonstrated that administration of low-dose leptin improved the transplant efficacy of ITx using a rodent diabetic model [113]. Lee et al. showed that leptin might enhance the therapeutic effect of ITx by reducing lipotoxicity, which impaired engraftment of intraportal transplanted islets in obese rats' abnormal leptin signaling [114]. On the other hand, adiponectin has anti-diabetic effects; it contributes to increased insulin sensitivity in target organs like muscle, liver, and endothelial cells [115] and plays anti-inflammatory and anti-atherogenic roles [116,117]. Therefore, adiponectin is a protective factor against obesity, type 2 diabetes, and cardiovascular diseases [118]. It is believed that the beneficial effects of adiponectin depend on its anti-inflammatory and angiogenic activities. Regarding its anti-inflammatory activity, Du et al. showed the therapeutic effect of adiponectin for transplanted islets by prevention of ischemic and reperfusion injury induced by activation of TNF-$\alpha$-induced nuclear transcription factor-$\kappa$B pathways [119]. Regarding its angiogenic activity, we showed that (1) vascularization of engrafted islets into white adipose tissue was prominent compared with engrafted islets in the renal subcapsular space and that (2) expressions of adhesion (Fn1, Itgb1, Itgb2) and angiogenic (Vegfa-c) factors in islets were enhanced by incubation of adiponectin (Sakata et al., Transplantation 2020, DOI: 10.1097/TP.0000000000003400; In press). These roles of adiponectin might contribute to the engraftment of islets into white adipose tissue.

### 3.2.3. Adipose Tissue Macrophage

Macrophage is characterized as an immune cell, which plays roles of phagocytosis and destruction of harmful organisms including bacteria, presents antigens to T cells and induces inflammation via

releasing cytokines. Regarding islet transplantation, Kupffer cells, liver tissue macrophages, contribute to prevention of islet engraftment. Kupfer cells are activated during IBMIR, which is occurred through complemental pathway in intraportal transplanted islets and attack to them by phagocytosis and secretion of inflammatory cytokines and free radicals [120]. On the other hand, the roles of adipose tissue macrophage against transplanted islets in white adipose tissue have not been fully discussed. Recently, Russo reviewed about the function of adipose tissue macrophages using lean and obese animal models. Adipose tissue macrophages are classified into to subtypes: tissue resident and monocyte-derived macrophages. While the population of adipose tissue macrophage is 10% of all cells in lean mice, they are increased to over 50% in obese mice [121,122]. It is considered that the 10% macrophages are tissue resident macrophages and increased macrophages are monocyte-derived macrophages. The tissue resident macrophages play as "M2" anti-inflammatory macrophages, which contribute to attenuation of inflammation by anti-inflammatory cytokines and regulation of immune cells in lean adipose tissue. On the other hand, in obese adipose tissue, monocyte-derived macrophages, which are characterized as "M1" macrophages, are increased and induce inflammation via activation of T cells and production of pro-inflammatory cytokines [123]. It is considered that the balance between tissue resident and monocyte-derived macrophages might influence of islet engraftment in white adipose tissue. However, further studies are necessary.

## 4. Clinical Trial of Islet Transplantation into White Adipose Tissue and Conclusion

Recent progress in clinical trials of islet transplantation using the omentum is seen in total pancreatectomy with islet autotransplantation. Stice et al. performed total pancreatectomy with islet autotransplantation into the omental pouch in chronic pancreatitis patients who could not receive complete intraportal infusion of islets due to portal hypertension and showed a similar glycemic control to "normal" intraportal islet autotransplantation [124]. Regarding allogeneic ITx, a phase I and II clinical trial for allogeneic islet cells transplanted onto the omentum, conducted by the University of Miami beginning in 2014 (ClinicalTrials.gov: NCT02213003), is ongoing. This trial aims to elucidate the therapeutic outcome of clinical ITx onto the omentum. The primary endpoint is achieving at least 6.5% HbA1c and no severe hypoglycemia by transplantation of at least 5000 IEQs/kg of body weight [125]. The primary outcomes will be cleared on May 2022. Another clinical trial from the University of Alberta was immaturely completed on November 2019 (NCT02821026) with no available results in published articles. The outcome of clinical intraomental islet transplantation is not fully discussed at this present moment. Further studies are needed.

Under this clinical trial, a case of laparoscopic ITx (602,395 IEQs/53.4 kg) onto the omentum was performed in a 43-year-old woman with type 1 DM who required more than 30 units of daily insulin and had episodes of unawareness and severe hypoglycemia [125]. Ameliorations in plasma C-peptide level, β cell function, and insulin sensitivity and resistance using the homeostasis model assessment index could be detectable 1 year after transplantation. This case report showed that severe metabolic condition can be recovered by intraomental ITx if sufficient numbers of islets can be implanted.

Finally, we touch on the issue of brown adipose tissue, which is the other type of adipose tissue located in the interscapular, supraclavicular, or para-aortic regions. Brown adipose tissue is a hypervascularized tissue [126] that contributes to maintaining body temperature. As a transplant site for ITx, brown adipose tissue has a unique rich vascularization that might support the engraftment of islets. On the other hand, the distribution of brown adipose tissue is limited, whereas white adipose tissue is located in the entire body. Recently, Xu et al. examined the transplant efficacy of ITx into white and brown adipose tissues. Their data did not show the superiority of brown adipose tissue compared with white adipose tissue in transplant efficacy [127].

## 5. Conclusions

The usefulness of white adipose tissue as a candidate transplant site for ITx has been proven in various animal studies. Clinical trials using white adipose tissue are ongoing.

**Author Contributions:** Conceptualization, N.S.; writing—original draft preparation, N.S.; writing—review and editing, G.Y.; supervision, S.K. All authors have read and agreed to the published version of the manuscript.

**Funding:** This study was funded by a Grant-in-Aid for Scientific Research (C) (grant number 19K09839, NS) from the Ministry of Education, Culture, Sports, Science, and Technology of Japan and an intramural grant from Fukuoka University.

**Conflicts of Interest:** The authors declare no conflict of interest.

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
