# Peer review of "White Adipose Tissue as a Site for Islet Transplantation"

_2673-3943, doi:10.3390/transplantology1020006_

Round 1

Reviewer 1 Report

Overall, very interesting review. Well written. All aspects covered. To date, published reviews are seldom in this area of basic and clinical research.

It is not clear, whether the study that is mentioned on line 297 is the one published by ref. 120 or by the authors of this review.

There are very few English style issues that could easily be corrected.

Author Response

Overall, very interesting review. Well written. All aspects covered. To date, published reviews are seldom in this area of basic and clinical research.

-Thank you.

It is not clear, whether the study that is mentioned on line 297 is the one published by ref. 120 or by the authors of this review.

-Thank you. Yes, this case report in Miami University (ref. 125 in revised version) was done under this clinical study (Allogeneic Islet Cells Transplanted onto the Omentum; ClinicalTrials.gov number, NCT02213003). For clarification of this relationship, I inserted the reference in the sentence (Page 12 Line 327).

There are very few English style issues that could easily be corrected.

-Thank you. I check and correct (e.g. “a large numbers of islets” to “a large number of islets” in Table 1).  

Reviewer 2 Report

The review by Sakata and coworkers outlines rationale for using white adipose tissue as a site for islet transplantation. The authors also provide a comparison with other known sites, such as kidney and hepatic portal circulation.

While generally informative, the authors should consider a discussion of the immune cell populations present within adipose tissue versus those in other sites (e.g., liver). For example, is there anything different about Kupffer cells versus adipose tissue macrophages that would make the white adipose tissue a preferred site? Or would the adipose tissue macrophages pose the same problems as the liver macrophage population?

Author Response

The review by Sakata and coworkers outlines rationale for using white adipose tissue as a site for islet transplantation. The authors also provide a comparison with other known sites, such as kidney and hepatic portal circulation.

While generally informative, the authors should consider a discussion of the immune cell populations present within adipose tissue versus those in other sites (e.g., liver). For example, is there anything different about Kupffer cells versus adipose tissue macrophages that would make the white adipose tissue a preferred site? Or would the adipose tissue macrophages pose the same problems as the liver macrophage population?

-Thank you. To the best of our knowledge, the influence of adipose tissue macrophage for islet engraftment has not been fully discussed. Thus, we discussed the characteristics of adipose tissue macrophages which are reviewed by Russo, et al (doi.org/10.1111/imm.13002). They are classified into two subtypes including tissue resident “M2” and monocyte derived “M1” macrophages. The former contributes to anti-inflammatory reaction in lean adipose tissue and the later induces inflammation in obese adipose tissue. We consider that the balance of these M1/M2 macrophages might influences the engraftment of islet in white adipose tissue. The discussion is inserted into the last paragraph of the third chapter (Page 11 Line 297 – Page 12 Line 317).

Reviewer 3 Report

The authors try to give an overview on Research performed on adipose tissue as Implantation site for islet Transplantation.

The Topic is interesting, however there is a lack of some studies which should be added to complete the Review such as

Creation of a prevascularized site for cell transplantation in rats.

Stiegler P, Matzi V, Pierer E, Hauser O, Schaffellner S, Renner H, Greilberger J, Aigner R, Maier A, Lackner C, Iberer F, Smolle-Jüttner FM, Tscheliessnigg K, Stadlbauer V.Stiegler P, et al. Xenotransplantation. 2010 Sep-Oct;17(5):379-90. doi: 10.1111/j.1399-3089.2010.00606.x.Xenotransplantation. 2010. PMID: 20955294.   There is a lot of Research performed on the creation of subcutaneous as well as intraomental Implantation sites for islet Transplantation wchich should be mentioned.   Moreover, the authors should provide the methodology of their literature Research and indicating how they Chose the articels mentioned in this paper.

Author Response

The authors try to give an overview on Research performed on adipose tissue as Implantation site for islet Transplantation.

The Topic is interesting, however there is a lack of some studies which should be added to complete the Review such as

Creation of a prevascularized site for cell transplantation in rats.

Stiegler P, Matzi V, Pierer E, Hauser O, Schaffellner S, Renner H, Greilberger J, Aigner R, Maier A, Lackner C, Iberer F, Smolle-Jüttner FM, Tscheliessnigg K, Stadlbauer V.Stiegler P, et al. Xenotransplantation. 2010 Sep-Oct;17(5):379-90. doi: 10.1111/j.1399-3089.2010.00606.x.Xenotransplantation. 2010. PMID: 20955294.  

-Thank you. Explanation of white adipose tissue (especially subcutaneous tissue) as a prevascularized site with reference is included in subcutaneous tissue section in the second chapter. (Page 8 Line 154-156)

There is a lot of Research performed on the creation of subcutaneous as well as intraomental Implantation sites for islet Transplantation which should be mentioned. Moreover, the authors should provide the methodology of their literature Research and indicating how they Chose the articles mentioned in this paper.

-Previously we reviewed about subcutaneous islet transplantation (doi: 10.1002/dmrr.2463), and thus, show the recent topic of subcutaneous islet transplantation in this review. (Page 8 Line 156-157)

Round 2

Reviewer 3 Report

The authors have adressed all comments and the Review seems ready for ublication now.